# Characteristics, Management, and Outcomes of Elderly Patients with Diabetes in a Covid-19 Unit: Lessons Learned from a Pilot Study

**DOI:** 10.3390/medicina57040341

**Published:** 2021-04-01

**Authors:** Marco Ricchio, Bruno Tassone, Maria Chiara Pelle, Maria Mazzitelli, Francesca Serapide, Paolo Fusco, Rosaria Lionello, Anna Cancelliere, Giada Procopio, Elena Lio, Enrico Maria Trecarichi, Carlo Torti, Concetta Irace

**Affiliations:** 1Azienda Ospedaliero-Universitaria Mater Domini, University Magna Graecia, 88100 Catanzaro, Italy; tassonebruno89@gmail.com (B.T.); chiarapelle87@gmail.com (M.C.P.); m.mazzitelli88@gmail.com (M.M.); francescaserapide@gmail.com (F.S.); paolofusco89@gmail.com (P.F.); rosarialionello0@gmail.com (R.L.); annacancelliere0@libero.it (A.C.); procopiogiada@libero.it (G.P.); elena.lio2409@gmail.com (E.L.); 2Department of Medical and Surgical Sciences, University Magna Graecia, 88100 Catanzaro, Italy; em.trecarichi@unicz.it (E.M.T.); torti@unicz.it (C.T.); 3Department of Health Science, University Magna Graecia, 88100 Catanzaro, Italy; irace@unicz.it

**Keywords:** Covid-19, diabetes, mortality, insulin therapy

## Abstract

*Background and objectives*: Diabetes may affect in-hospital mortality of patients with Coronavirus disease 2019 (COVID-19). We have retrospectively evaluated clinical characteristics, diabetes management, and outcomes in a sample of COVID-19 patients with diabetes admitted to our hospital. *Materials and Methods*: All patients admitted to the Infectious Diseases Unit from 28 March 2020, to 16 June 2020, were enrolled. Clinical information and biochemical parameters were collected at the time of admission. Patients were ranked according to diabetes and death. *Results*: Sixty-one patients with COVID-19 were analyzed. Most of them were from a long-term health care facility. Mean age was 77 ± 16 years, and 19 had type 2 diabetes (T2D). Eighteen patients died, including 8 with T2D and 10 without T2D (*p* = 0.15). Patients with diabetes were significantly older, had a higher prevalence of cardiovascular diseases, and a significantly lower lymphocyte count. No significant relationship was found between diabetes and in-hospital mortality (Odds Ratio OR 2.3; Confidence Interval CI 0.73–7.38, *p* = 0.15). Patients with diabetes were treated with insulin titration algorithm. Severe hypoglycemic events, ketoacidosis and hyperosmolar hyperglycemias did not occur during hospitalization. Mean pre-meal capillary blood glucose was 157 ± 45 mg/dL, and the coefficient of variation of glycaemia was 29%. *Conclusions*: Our study, albeit limited by the small number of subjects, did not describe any significant association between T2D diabetes and mortality. Clinical characteristics of patients, and acceptable glucose control prior and during hospitalization may have influenced the result. The use of an insulin titration algorithm should be pursued during hospitalization.

## 1. Introduction

The prevalence of diabetes in CoronaVirus Disease-2019 (COVID-19) hospitalized patients is similar to the overall prevalence of diabetes in the general population. Chinese researchers reported a prevalence of diabetes in COVID-19 patients ranging from 5.3 to 19.5%, compared to a prevalence of 11% in the general population [1]. The US reports are quite similar, with prevalence of diabetes in COVID-19 patients of 11%, and the overall prevalence in the population of 13% [2]. In Italy, the prevalence of diabetes was demonstrated to be slightly lower in infected patients (9%), but again similar to the prevalence of the general population (11%) [3]. Although diabetes cannot be considered a risk factor of COVID-19 infection, it may be responsible for worse outcomes such as death and mechanical ventilation [4]. Unfavorable prognosis due to diabetes has also been described during MERS (Middle East Respiratory Syndrome) and SARS (Severe Acute Respiratory Syndrome) CoV-1 epidemic [5,6,7]. The relationship between diabetes, death, and admission into the Intensive Care Unit (ICU) seems to be mediated by comorbidities, which were more prevalent in patients with diabetes, such as cardiovascular and cerebrovascular diseases, chronic kidney disease, and hypertension [8]. The relationship between glycated hemoglobin (HbA1c) at the time of hospital admission and clinical outcomes seems to be controversial. Indeed, one study demonstrated a greater risk of death among those with poor glycemic control [9], while the French nationwide CORONADO study did not find any significant associations between HbA1c, death and mechanical ventilation within seven days after hospitalization [10]. Conversely, hyperglycemia at the time of the admission, in patients with and without prior diabetes, seems a strong predictor of worse outcomes [11,12].

Appropriate management of diabetes during hospitalization is crucial to provide better outcomes, and to reduce length of hospitalization. Insulin therapy should be preferred to other non-insulin treatments, as suggested by the American Diabetes Association [13]. Experts recommend a feasible and structured insulin regimen, and a target glucose range of 140–180 mg/dL. Recently, it is growing the interest in glycemic variability, which seems to be an independent risk factor for morbidity and mortality in hospitalized patients [14].

In the present research, we have retrospectively evaluated clinical characteristics, diabetes management, and outcomes in a sample of COVID-19 patients with type 2 diabetes (T2D) admitted to our hospital in order to assess whether an insulin titration algorithm was able to maintain glycaemia into a desirable range.

## 2. Materials and Methods

The research is a retrospective study including all patients admitted to the Infectious Diseases Unit at the “Mater Domini”, Teaching Hospital, University Magna Graecia, Catanzaro, Italy from 28 March to 16 June 2020. Almost all were elderly patients from long-term health care facility (LTHCF). Clinical and biochemical parameters [fasting plasma glucose, lipids, Interleukin-6 (IL6), C-Reactive Protein (CRP), fibrinogen, ferritin, alanine amino transaminase (ALT), aspartate amino transaminase (AST), γ-glutamyl-transferase (γGT), lactate dehydrogenase (LDH), creatinine, lymphocytes count, platelet count (PLT), D-dimer, sodium, total cholesterol, HDL-cholesterol, triglycerides], were collected at the time of the admission into the hospital, while the presence of prior T2D, hypertension, cardiovascular disease (CVD), chronic obstructive pulmonary disease (COPD), cerebrovascular disease, psychiatric and neurological disorder, and ongoing treatments were collected from clinical records provided by the LTHCF.

At the time of the admission, non-insulin hypoglycemic treatment was discontinued, while ongoing basal-bolus insulin therapy was confirmed. In insulin naïve patients, a starting safe basal-bolus treatment (long-acting insulin 10 U, and rapid-acting insulin 10 U) was suggested. Rapid-acting insulin dose was injected before meal and adjusted daily according to the pre-meal capillary blood glucose (Table 1).

Long-acting insulin was injected at bed-time and the dose was adjusted every 1–2 days according to fasting plasma glucose or pre-breakfast glycemia (Table 2).

In case of persistent high blood glucose level (>250 mg/dL) despite insulin titration, intravenous insulin injection was suggested.

Statistical analyses were performed by IBM-SPSS Statistics v.23 (IBM, New York, US). Patients were divided into two groups according to the presence or absence of diabetes. Subjects with diabetes were further divided into deceased and not-deceased patients. Variables not normally distributed were: plasma glucose, lipids, IL6, CRP, fibrinogen, ferritin, ALT, AST, γGT, LDH creatinine, lymphocytes count, D-dimer, sodium, total cholesterol, HDL-cholesterol, and triglycerides. Data have been expressed as mean ± standard deviation (SD), and percentage. Differences between groups were evaluated by *t*-test and Mann Whitney U test for unpaired data and chi-square test. Univariate regression analysis was performed to assess the relationship between diabetes and in-hospital mortality. The multiple logistic regression analysis was performed to evaluate the independent association between death and variables significantly different between patients with and without diabetes. Pre-meal capillary blood glucose values of each patient measured during hospitalization at each insulin injection were grouped and mean ± SD was calculated. Only patients for whom insulin was titrated for at least 5 days were included in the analysis. Glycemic variability was calculated as the coefficient of variation (CV) using the following formula: (Standard deviation/mean blood glucose) × 100.

A *p*-value of less than 0.05 was considered to be statistically significant.

## 3. Results

A total of 61 patients were enrolled in the study. Nineteen patients out of 61 (31%) had T2D. No patients had diabetic ketoacidosis or hyperosmolar hyperglycemia state at the admission or during hospitalization or severe hypoglycaemia. As far as COVID-19 treatment is concerned, a total of 46 out of 61 patients received combination therapy with hydroxychloroquine plus azithromycin according to the study protocol by Gautret et al. [15], followed by electrocardiogram (ECG) monitoring. No patients had cardiac complications. Based on clinical judgment, in 30 patients out of 61, methylprednisolone 40 mg twice daily was administered. All patients received enoxaparin in the absence of low platelet count or alternatively fondaparinux. Three patients received as add-on therapy subcutaneous tocilizumab. None of the patients was admitted into the ICU during the hospitalization.

Clinical characteristics and biochemical parameters of all patients and patients divided according to the presence or absence of T2D are displayed in Table 3. Creatinine and fasting blood glucose were the only variables significantly different between patients with and without diabetes.

Eighteen patients (29.5%) died, among whom 8 (44%) with T2D and 10 (56%) without T2D (*p* = 0.15). The mean time from admission to in-hospital mortality was 14 ± 11 days in patients without diabetes and 9 ± 6 days in T2D patients (*p* = 0.46). Patients with T2D were further divided according to in-hospital mortality, and characteristics are described in Table 4. Patients who died were significantly older, had a higher prevalence of prior CVD, and a significantly lower lymphocytes count. No significant relationship was found between diabetes and in-hospital mortality (OR: 2.3; 95% CI 0.73–7.38, *p* = 0.15). Age was the only variable independently associated with death in all patients: OR 1.07; 95% CI 1.00–1.15, *p* = 0.05. The result was not confirmed when patients were divided between those with and without diabetes.

The basal and rapid-acting insulin dose was titrated at each meal and at bed-time according to the titration algorithms in 9 patients with diabetes. The mean daily capillary pre-meal blood glucose values has been displayed in Figure 1. For one patient the algoritm was used for 5 days, for a second one for 7 days, because one was discharged and the other one did not further require insulin treatment. For the all others 7 patients the algorithm was used for more than 2 weeks. Blood glucose data of the remaining 2 non-deceased patients have not been displayed because they did not receive insulin during hospitalization due to acceptable blood glucose values. None of the nine patients who were managed by the titration algorithm died. The mean and standard deviation of all pre-meal capillary blood glucose from the hospitalization until discharge, in insulin-treated patients, was 157 ± 45 mg/dL, and the CV was 29%. Among diseased patients with diabetes the algorithm was not used for the following reasons: 3 patients died in the first days of hospitalization, 5 had capillary blood glucose values lower than the target without insulin, and 1 patient was managed during hospitalization with intravenous insulin.

## 4. Discussion

The present study suggests that diabetes does not increase the risk of death in COVID-19 patients in elderly patients with COVID-19. This finding is in line with the results of other studies [16]. More in details, the adequate controlled diabetes at the time of admission and during hospitalization might reduce the risk of in-hospital mortality associated with T2D. Conversely, other studies support the hypothesis that diabetes is *di per se* a risk factor for mortality [17]. Zhu et al. [18] have recently described, in a retrospective longitudinal study, that well-controlled blood glucose (blood glucose values within 70 to 180 mg/dL) is associated with markedly lower mortality compared to individuals with poorly controlled blood glucose (blood glucose values exceeding 180 mg/dL).

Glucose control during hospitalization, in our study, appeared to be appropriate both for the mean pre-meal blood glucose value and for CV. Indeed, both the mean and the CV were respectively whithin and under the threshold value suggested by the international consensus on new glucometric parameters [19]. The use of the insulin titration algorithm might have influenced glycemic control. Furthermore, none of the patients managed with the titration algorithm had severe hypoglycaemia or diabetic ketoacidosis (DKA) during hospitalization. Again, the absence of hypoglycaemic events might have influenced the result of the study. Hypoglycaemia has been associated with increased mortality in hospitalized patients [20]. Our study for the first time has evaluated the glycemic control over time in hospitalized patients with diabetes and COVID-19. Other studies conducted so far have only evaluated the impact of blood glucose at the time of hospitalization on the mortality [21]. Our study is however limited by the small number of patients managed with the algorithm and therefore by the lack of an appropriate survival analysis.

Among variables which may impact on glucose levels, we have to consider the concomitant medication. More than 2/3 patients in our group received hydroxychloroquine and approximately half of them received corticosteroids. Hydroxychloroquine may lower blood glucose and corticosteroids may raise it, so further studies in COVID-19 patients should consider their effects. Also, in clinical practice, the possible effects of these drugs on glucose control should be considered, with pre-prandial glycaemia measured even in non-diabetic patients taking these drugs.

The benefit of a strict monitoring, should be balanced, however, against the risk of infections for the healthcare workers. For this reason, we prioritized to diabetic patients or those symptomatic for hypo or hyperglycaemia, while the other patients were not monitored.

An alternative interpretation for the lack of an association between T2D and mortality could be that possible pre-existing complications of T2D are more important. In apparent support to this interpretation, in our previous analysis [16], CV diseases (and not T2D) emerged as an independent predictor of death over T2D per se. More in general, CVD is a predictor of mortality in infected patients regardless of diabetes and even in non COVID-19 patients [22]. Along the same line, Apicella et al. [23] showed that, in COVID-19 patients, poorer prognosis of people with diabetes is likely to be the consequence of a syndromic nature of diabetes, in which hyperglycaemia, hypertension, obesity, and CVD all contribute to increase the risk of death. Therefore, albeit TD2 *per se* may not emerge as a risk factor, attention should be given to these patients who may be more fragile for comorbidities, especially the cardiovascular one.

We found that patients with diabetes who died were older and had lower lymphocyte count than patients with diabetes who survived. The recent meta-analysis by Huang et al. [24] has demonstrated the strong relationship between diabetes and poor COVID-19 outcomes but not with ICU admission. The authors have also demonstrated that the relationship between diabetes and worst infection outcomes is complex and affected by the prevalence of other comorbidities as hypertension. In our sample the prevalence of hypertension was comparable between patients with and without diabetes as well as between deceased and survivors with diabetes.

Our study is affected by several limitations. First, the small number of patients may have precluded to get a definitive answer as to whether or not T2D *per se* is a variable independently correlated or predictive of death in COVID-19 patients. Second, unfortunately, we did not have a recent glycated haemoglobin value and could not measure it during hospitalization. Without HbA1c it is impossible to separate undiagnosed T2D (for example, HbA1c 11% and casual plasma glucose 250 mg/dL) and new onset diabetes after COVID-19 infection (for example, HbA1c 5.6% and casual plasma glucose 260 mg/dL). Notwithstanding these limitations, our study has strengths. First, as far as age is concerned, population was quite homogeneous. Second, almost all (50 patients) came from LTHCF where a COVID-19 outbreak occurred, so they were infected almost at the same time, making the analysis of intra hospital mortality more accurate and not biased by different length of infection prior to enrollment. Lastly, our operational study evaluated, for the first time, titration algorithm which appeared to be beneficial in allowing a good glucose control. At the same time, our protocol was designed with the aim of reducing the number of contacts between healthcare workers and infecting patients, thus allowing to contain the intra hospital risk of infection.

## 5. Conclusions

In conclusion, mortality rate in elderly patients with COVID-19 may not be affected by diabetes to a significant extent, provided that glycemic control is acceptable both at admission and during hospitalization, avoiding severe hypoglycemic events, and concomitant with optimized management of complications. Our protocol for management of diabetes could also help control the intra-hospital risk of SARS-CoV-2 infection. More studies are needed to validate these results.

## Figures and Tables

**Figure 1 medicina-57-00341-f001:**
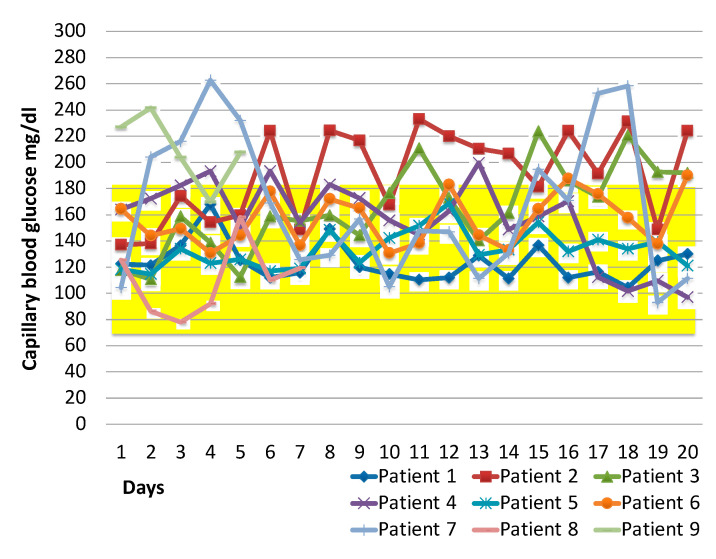
Blood glucose trend during hospitalization. Each point represents the daily mean fasting pre-meal blood glucose. The highlighted area represents the optimal blood glucose range during the hospitalization.

**Table 1 medicina-57-00341-t001:** Rapid-acting insulin titration algorithm according to pre-meal fasting glycemia.

Pre-Meal Blood Glucose (mg/dL)	Rapid-Acting Insulin (Unit, U)
60–69	Start meal and inject half units within 20 min after meal
70–79	−3 U
80–99	−2 U
100–139	−1 U
140–160	No change
160–180	+1 U
181–200	+2 U
201–250	+3 U
251–300	+4 U
301–350	+6 U delay meal (1/2 h)
>350	+8 U delay meal (1/2 h)

**Table 2 medicina-57-00341-t002:** Long-acting insulin titration algorithm according to fasting or pre-breakfast glycemia.

Fasting or Pre-Breakfast Blood Glucose (mg/dL)	Long-Acting Insulin (Unit, U)
>140	+ 2 U
139–110	No change
<110	−2 U

**Table 3 medicina-57-00341-t003:** Clinical and biochemical characteristics of COVID-19 patients grouped and divided according to the presence or absence of diabetes.

	All Patients (*n* = 61)	Patient with T2D (*n* = 19)	Patient without T2D (*n* = 42)
Age, years	77 ± 16	81 ± 16	75 ± 15
Male, %	48	47	48
FPG, mg/dL	119 ± 53	157 ± 77	102 ± 19 *
AST, U/L	56 ± 155	94 ± 272	37 ± 21
ALT, U/L	54 ± 208	111 ± 371	36 ± 22
γGT, U/L	45 ± 55	53 ± 82	41 ± 38
Creatinine, mg/dL	1.05 ± 0.5	1.17 ± 0.5	0.99 ± 0.5 *
Fibrinogen, mg/dL	434 ± 112	426 ± 129	438 ± 105
Ferritin, ng/mL	656 ± 801	750 ± 1149	614 ± 598
CRP, mg/L	56 ± 75	60 ± 69	54 ± 78
LDH, mU/mL	610 ± 270	632 ± 225	600 ± 290
Il-6, pg/mL	58 ± 102	68 ± 84	53 ± 110
PLT, ×10^3^/uL	210 ± 87	183 ± 75	222 ± 90
Lymphocytes count, ×10^3^/uL	1.28 ± 0.7	1.08 ± 0.5	1.36 ± 0.7
D-dimer, mg/l	2.2 ± 4.2	2.7 ± 4.6	2 ± 4
Sodium, mmol/L	140 ± 18	143 ± 7	138 ± 21
Total Cholesterol, mg/dL	153 ± 36	145 ± 37	156 ± 36
HDL-cholesterol, mg/dL	34 ± 9	32 ± 7	35 ± 10
Triglycerides, mg/dL	134 ± 58	147 ± 65	129 ± 54
CVD (%)	32	31	32
Neurological disorder, %	43	42	44
Hypertension, %	68	78	63
Psychiatric disorder, %	25	37	20
COPD, %	15	21	12

Data are expressed as mean ± SD and percentage. T2D: type 2 diabetes; FPG: fasting plasma glucose; AST: aspartate aminotransferase; ALT: alanine aminotransferase; γGT: gamma-glutamyl transferase; CRP: C-reactive protein; LDH: lactate dehydrogenase; IL-6: interleukin 6; PLT: platelet; CVD: cardiovascular disease; COPD: chronic obstructive pulmonary disease. * *p* < 0.05 vs. with diabetes.

**Table 4 medicina-57-00341-t004:** Clinical and biochemical characteristics of deceased and non-deseased COVID-19 patients with diabetes.

	Deceased (*n* = 8)	Non-Deceased (*n* = 11)
Age, years	90 ± 8 *	74 ± 18
Glycaemia, mg/dL	159 ± 86	155 ± 74
AST, U/L	185 ± 418	28 ± 11
ALT, U/L	257 ± 592	18 ± 10
GGT, U/L	70 ± 119	41 ± 41
Creatinine, mg/dL	1.32 ± 0.6	1.06 ± 0.3
Fibrinogen, mg/dL	388 ± 160	454 ± 100
Ferritin, ng/mL	1045 ±1875	562 ± 194
CRP, mg/L	94 ± 91	35 ± 35
LDH, mU/mL	690 ± 235)	596 ± 233
Il-6, pg/mL	107 ± 119	39 ± 25
PLT, ×10^3^/uL	147 ± 30	209 ± 87
Lymphocytes count, ×10^3^/uL	0.72 ± 0.3 *	1.3 ± 0.5
D-dimer, mg/l	4.8 ± 6.7	1.1 ± 0.7
Sodium, mmol/L	147 ± 6	141 ± 6
Total Cholesterol, mg/dL	141 ± 56	148 ± 24
HDL-cholesterol, mg/dL	31 ± 8	33 ± 7
Triglycerides, mg/dL	157 ± 101	143 ± 42
CVD, %	62 *	9
Neurological disorder, %	50	36
Hypertension, %	75	82
Psychiatric disorder, %	50	27
COPD, %	25	18

Data are expressed as mean ± SD and percentage. FPG: fasting plasma glucose; AST: aspartate aminotransferase; ALT: alanine aminotransferase; γGT: gamma-glutamyl transferase; CRP: C-reactive protein; LDH: lactate dehydrogenase; IL-6: interleukin 6; PLT: platelet; CVD: cardiovascular disease; COPD: chronic obstructive pulmonary disease. * *p* < 0.01 vs. non-deceased.

## Data Availability

The datasets generated and/or analysed during the current study are not publicly available due to privacy and presence of personal data but are available from the corresponding author on reasonable request.

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
