# Peer review of "Characteristics, Management, and Outcomes of Elderly Patients with Diabetes in a Covid-19 Unit: Lessons Learned from a Pilot Study"

_medicina, 2021, doi:10.3390/medicina57040341_

Round 1
Reviewer 1 Report
The paper analyzes the clinical characteristics, diabetes abetes management, and outcomes in a group of COVID-19 patients with T2DM. The intorudtion clearly states the purpose of the study, the methodology is adequate, the reults need to be futher fetailed, what other treatment did the patient recieve? did they recieve anticoagulants? did they recieve dexamethazone? please further detail. Please make a muliple regression analysis in the total group and in both diabtic and non-diabetic groups to see which are the risk factors for death. The graphical part needs to be gratly improved. The conclusion support the objectives however the results part needs to be improved and futher detailed in order to give strength to your conclusions.please improve the tables form, they look very negligent done, please improve the graphical part the graphic is not very inteligible, please further discuss your paper compared to other results, is the good glycemic control an explanation for your results?
Author Response
The paper analyzes the clinical characteristics, diabetes abetes management, and outcomes in a group of COVID-19 patients with T2DM. The introduction clearly states the purpose of the study, the methodology is adequate, the results need to be further detailed, what other treatment did the patient receive? did they receive anticoagulants? did they receive dexamethazone? please further detail. Please make a multiple regression analysis in the total group and in both diabetic and non-diabetic groups to see which are the risk factors for death. The graphical part needs to be gratly improved. The conclusions support the objectives however the results part needs to be improved and further detailed in order to give strength to your conclusions. Please improve the tables form, they look very negligent done, please improve the graphical part the graphic is not very inteligible, please further discuss your paper compared to other results, is the good glycemic control an explanation for your results?
We thank the reviewer for his general comment and suggestions.
- Methylprednisolone 40 mg twice daily was administrated to 30 patients over 61. All patients received Enoxaparin in the absence of low platelet count or fondaparinux. This information has been added to the main manuscript in the result section (Results, Page 5, line 130-132).
- As suggested by the reviewer, we have included an additional multiple regression analysis to evaluate the variables independently associated with death in patients with and without diabetes and in the overall group. The results have been added in the main text (Results, page 6, line 167, page 7 lines 168-170). The analysis has also been described in the Material and Methods section (Page 5, line 114-116)
- Table and figures have been modified (according to the instruction the tables have been included into the main text, and for this reason in some case the table is included in two different pages).
Reviewer 2 Report
The authors evaluate retrospectively clinical characteristics in type 2 diabetic patients affected of COVID-19 as well as the insulin titration algorithm along 20 days.
Authors have submitted a version of the manuscript with important errors, which show sloppy work.
The following are those mistakes that need to be resolved
Page 5 Line 125. Importantly, Table 4 must be included to the manuscript
Page 5 Lines 129-136. Paragraph must be improved. If insulin titration was realized in T2DM patients, the authors should revise the number of patients indicated in the Figure 1. Authors explain that from the 61 participants in the study, 19 were diagnosed with type 2diabetes, and of these, 8 died during the course of the study (remaining 11 participants). They indicate nine patients included in figure 1, for this reason, the remaining patients not included in Figure 1 should be 2, not 9
Page 6 Figure 1. An explanation about patient 9 and 8 are not evaluated for the total period of time should be added by the authors
Page 8 Lines 223- 229 Conclusions are not corresponding to this study.
Other considerations:
Page 1 Lines, 24 and 30, authors should reconsider to change “insulin algorithm titration algorithm” for “insulin titration algorithm” along the manuscript
Page 1 Line 44: Authors should indicate MERs and SARS meaning.
All tables should improve their appearance, so that they are included on the same page, and not split into two pages. In addition, table 3 needs a table footnote with abbreviations and AST values should be revised for Total participants column. In addition, groups compared for statistical analysis should be added
Cardiovascular disease abbreviation should be revised along the manuscript
Author Response
The authors evaluate retrospectively clinical characteristics in type 2 diabetic patients affected of COVID-19 as well as the insulin titration algorithm along 20 days.
Authors have submitted a version of the manuscript with important errors, which show sloppy work. The following are those mistakes that need to be resolved:
We thank the reviewer to underlie the errors of the manuscript. We really apologize, but we believe that something was wrong during the submission.
Page 5 Line 125. Importantly, Table 4 must be included to the manuscript.
Table 4 has now been included into the main text.
Page 5 Lines 129-136. Paragraph must be improved. If insulin titration was realized in T2DM patients, the authors should revise the number of patients indicated in the Figure 1. Authors explain that from the 61 participants in the study, 19 were diagnosed with type 2diabetes, and of these, 8 died during the course of the study (remaining 11 participants). They indicate nine patients included in figure 1, for this reason, the remaining patients not included in Figure 1 should be 2, not 9. Page 6 Figure 1. An explanation about patient 9 and 8 are not evaluated for the total period of time should be added by the authors
We have changed the comment to Figure 1 (Page 8, lines 181-193). Indeed, we have now explained the use of the algorithm in survived patients with diabetes. We have also better explained the use of the algorithm in the Material and Methods. (Page 3, Lines 86-90; Page 4 lines 91-106)
Page 8 Lines 223- 229 Conclusions are not corresponding to this study.
We are very surprised for this inconvenient. That was of course not our conclusions. Again, we believe that an error occurred in the system. We have deleted this paragraph. Furthermore, according to the instruction for the authors, a conclusion section is not required.
Other considerations:
Page 1 Lines, 24 and 30, authors should reconsider to change “insulin algorithm titration algorithm” for “insulin titration algorithm” along the manuscript.
We have corrected the misprint.
Page 1 Line 44: Authors should indicate MERs and SARS meaning.
We have clarifies the meaning of MERS and SARS.
All tables should improve their appearance, so that they are included on the same page, and not split into two pages. In addition, table 3 needs a table footnote with abbreviations and AST values should be revised for Total participants column. In addition, groups compared for statistical analysis should be added
Cardiovascular diseases abbreviation should be revised along the manuscript
We have improved the appearance of the tables so that they are now included in a single page and a table footnote with abbreviations has been included in Table 3 and Table 4. The value of AST has been revised and adjusted. We have specified the comparison between groups in Table 3 (Table and text, Result section, Page 6, line 154-157). The abbreviation for cardiovascular disease has been revised and modified.
Reviewer 3 Report
Although if find the idea of interest this study is obviously underpowered to detect any significant association between T2D diabetes and mortality.
Therefore I urge the authors to extend this retrospective study from June 16th, 2020 and instead rerun the analysis from today in order to increase power.
Author Response
Although if find the idea of interest this study is obviously underpowered to detect any significant association between T2D diabetes and mortality.
Therefore I urge the authors to extend this retrospective study from June 16th, 2020 and instead rerun the analysis from today in order to increase power.
We thank the reviewer for his comment.
The observation is definitely appropriate. We have highlighted that in the discussion. However, we believe that the inclusion of patients of the second wave might represent a bias. Indeed, as you can image, the overall strategies managing patients with COVID-19 is partially changed.
Nevertheless, accordingly to the reviewer comment, we have changed the title of the manuscript as follows: Characteristics, management, and outcomes of elderly patients with diabetes in a covid-19 unit: lessons learned from a pilot study.
Round 2
Reviewer 2 Report
Despite the fact that in the previous revision I considered that the manuscript had serious errors, and for this reason it should be rejected. After this second revision, I have found that the authors have fixed the mistakes they made.
I understand that they uploaded an incorrect previous version, since the system cannot remove tables from the text or change paragraphs of conclusions.
In spite of all this, and given that the publisher has sent me an email requesting a new revision. I have done it as if it were the first time I had read the manuscript. Really, they have improved the second version.
Therefore, this time my conclusion is that the article is ready to be published after a few small changes that, I suggest the authors make to increase the quality of the article in order to be more understandable to the audience.
- Page 4 Line 126. Authors should explain the ECG abbreviation
- Page 4 Table 3. Authors must check the way they write the results: mean ± SD, or mean±SD, but always the same.
- Page 5 Table 4. The same as above. Consistency along the manuscript
- Page 9 Line 201. “Indeed, the both…”, “the” is not necessary
- Page 9 Line 205 DKA should be explained
- Page 10 Line 232. CV diseases must be CVDs because CV has a different meaning in this manuscript, in fact coefficient of variation at page 3 line 111
Reviewer 3 Report
Even though the authors have changed the title stating that this is a pilot study is still think that has not enough scientific potential to merit publication in Medicina because of the few participants included in the study.